# An Advanced Data Fusion Method to Improve Wetland Classification Using Multi-Source Remotely Sensed Data

**DOI:** 10.3390/s22228942

**Published:** 2022-11-18

**Authors:** Aaron Judah, Baoxin Hu

**Affiliations:** Department of Earth and Space Science and Engineering, York University, Toronto, ON M3J 1P3, Canada

**Keywords:** wetlands, multi-source, data fusion, Dempster–Shafer theory, random forest, ensemble classifier

## Abstract

The goal of this research was to improve wetland classification by fully exploiting multi-source remotely sensed data. Three distinct classifiers were designed to distinguish individual or compound wetland categories using random forest (RF) classification. They were determined, in part, to best use the available remotely sensed features in order to maximize that information and to maximize classification accuracy. The results from these classifiers were integrated according to Dempster–Shafer theory (D–S theory). The developed method was tested on data collected from a study area in Northern Alberta, Canada. The data utilized were Landsat-8 and Sentinel-2 (multi-spectral), Sentinel-1 (synthetic aperture radar—SAR), and digital elevation model (DEM). Classification of fen, bog, marsh, swamps, and upland resulted in an overall accuracy of 0.93 using the proposed methodology, an improvement of 5% when compared to a traditional classification method based on the aggregated features from these data sources. It was noted that, with the traditional method, some pixels were misclassified with a high level of confidence (>85%). Such misclassification was significantly reduced (by ~10%) by the proposed method. Results also showed that some features important in separating compound wetland classes were not considered important using the traditional method based on the RF feature selection mechanism. When used in the proposed method, these features increased the classification accuracy, which demonstrated that the proposed method provided an effective means to fully employ available data to improve wetland classification.

## 1. Introduction

Wetlands are critically linked to major issues such as climate change, wildlife habitat health, biodiversity, and groundwater issues. More specifically, wetlands play important roles in flood mitigation, water quality protection, and global carbon and methane cycles, acting as buffers against droughts, protecting coastlines from rising tides and storms, and being responsible for sediment retention [1,2,3,4,5]. North American and global wetland losses are estimated to be on the order of 50% since the early 1700s, and nearly 35% of global wetlands have been lost since 1970 [2,3,4,5]. Wetland conservation is well established as a matter of national and international public policy. Accurate maps of wetland boundaries and their changes are essential for effective monitoring, and remotely sensed imagery provides researchers with a means to achieve those goals [6,7,8,9,10,11,12,13,14,15,16,17,18,19,20,21].

Remotely sensed imagery has been used to generate wetland maps with various levels of success [6,7,8,9,10,11,12,13,14,15,16,17,18,19,20,21,22]. High-spatial-resolution remotely sensed imagery has created some of the most accurate wetland maps with the disadvantages of limited coverage and large time and resource demands; turnaround times for these products can be years [4]. Wetland classification using medium-spatial-resolution satellite imagery such as the Landsat or ASTER series of sensors is common and considered a standard approach [17], with the best results found when one class dominates the classification area (>30 m^2^) [17]. However, studies have shown that, when mixtures of wetland types are of the same order as the sensor resolution [17], class separation becomes more difficult. The addition of ancillary data such as elevation maps and field samples to the classification process of medium resolution imagery has been found to increase classification accuracies, with accuracies ranging from 30% to 82%, depending on the techniques used [12,17,23,24,25].

In most classification publications, only accuracy-related measures were reported. However, it is also important to examine the uncertainty of the classification results and the nature of the misclassification [26]. It is common that the uncertainty is high for some of the misclassified pixels. However, for some misclassified pixels, the uncertainty might be low, which is more problematic in the utilization of the classification results. The same is true for the correctly classified pixels but with high uncertainty. In our previous studies [27,28], where a random forest (RF) classifier was executed on a concatenating set of features derived from multi-source remotely sensed data, up to 30% of misclassified pixels were classified with a confidence level of greater than 85%. The results also showed that, for about 40% of the misclassified pixels, the correct class (ground truth) was found to be the second choice, and, in some cases, the difference in the posterior probability between the top two categories was only ~0.05. The method used in our previous studies [27,28] is commonly used for cover type classification using multi-source remotely sensed data, as discussed more in detail. An apparent conclusion is that the aggregated features may not be able to reliably separate the land-cover types of interest. However, this does not mean that these cover types cannot be separated by those features. With the aggregation, some features important to separate certain cover types might be masked by others. Classification may be improved if features are utilized differently. In that vein, one motivation for this study was to develop advanced methods to reduce these errors by maximizing the usage of available datasets. 

With the wide availability of multi-source remotely sensed data, data fusion techniques have also been utilized to improve the classification of remotely sensed imagery [29,30,31,32,33,34,35,36]. Generally speaking, data fusion can be performed at the pixel, feature, and decision levels. With pixel-level fusion, an improved fused image is generated by combining two data sources [37]. The most notable example is the pan-sharpened image by combining the low-spatial-resolution multi-spectral imagery and the high-spatial resolution panchromatic imagery. In this context of image classification, individual data sources are not explicitly analyzed and potentially not fully utilized with the pixel-level fusion methods. Accordingly, data fusion at the feature level and decision level is the focus of this study. Feature-level fusion involves concatenating sets of features before the classification process. Decision-level fusion involves the merging of decisions from multiple classifiers either with different features or using different classification methods. The feature-level fusion is the most commonly used, due to its simplicity and demonstrated success [31,38]. However, the high dimensionality in the feature space that results from feature-level fusion, even after feature reduction efforts, is likely to be a concern for applications where the size of training samples is small [39]. In addition, features derived from different data sources are usually treated equally by most classifiers (such as RF methods), even though some of the data sources may be more reliable than others [26]. On the contrary, each data source is analyzed separately in decision-level fusion, and the uncertainty and imprecision associated with each data source can be measured and considered in the fusion process. The challenge with decision-level fusion lies in the selection of propositions for each data source and effective ways to combine the decisions. In this study, we developed an ‘ensemble classifier’ based on both feature-level and decision-level fusion to improve wetland classification, henceforth referred to as the ensemble classifier. Below, we describe the motivation for the method development and its uniqueness compared with existing fusion methods for remote sensing classification.

A previous study of ours showed that broad class separations are an effective way of classifying data in a hierarchical fashion [28]. This study showed that different image features and/or datasets could be tailored, through analysis, to be used at different stages of classification, within a hierarchy, in order to create superior or more consistent results, when compared to previous studies which have relied strictly on the resolution or characteristics of those inputs to drive splits in those classification hierarchies [40,41,42,43,44,45,46,47,48,49,50,51]. In the current study, we leveraged these broad class separations to create two additional classifiers, in addition to a traditional classifier focused on separating individual classes, to create an ensemble classifier to best utilize the available datasets for the study area. This was intended to not only increase classification accuracy but also reduce the number of high-confidence misclassified pixels through additional observation and analysis (as mentioned earlier). We propose to combine these classifiers through a Dempster–Shafer (D–S) theory of combined evidence using the results from the three classifiers, due to its capability of handling uncertainty [32]. 

This study is unique in two aspects considering the proposed method and analysis. For most existing methods related to ensemble classifiers and/or decision-level fusion, multiple classifiers are employed to deal with identical sets of classes [30,34,52,53]. Our work fully leveraged the diversified features derived from available multi-source remotely sensed data to be used in individually designed classifiers with unique class propositions. Furthermore, prior knowledge on wetland cover types and remotely sensed data used was also utilized in the selection of features for each classifier, in addition to the data-driven machine learning approaches that were commonly and exclusively used in most classification methods. One might argue that hierarchical classification methods effectively utilized features to separate different categories of cover types. However, the uncertainty associated with the classifiers in the hierarchy was not addressed [28] and was difficult to be accounted for in the lower parts of the hierarchy. The D–S theory used in this study provides an effective means to consider the uncertainty in each classifier. Similarly, to the argument on rule-based past-processing applied to classification maps, the advantage of this method in in its handling of uncertainty and avoiding the selection of thresholds in any rule-based methods. In this study, detailed analysis of the nature of misclassification was also carried out, which was lacking in the literature [29,30,34,52,53].

It is worth mentioning that deep learning is attracting substantial attention in cover type classification using multi-source remotely sensed data including but not limited to wetland classification [40,41,42,43,44,45,46]. These developed deep learning methods can also be categorized as pixel-level, feature-level, and decision-level fusion. Most of them implement feature-level fusion. The issues discussed earlier on pixel-level and feature-level fusion for classification also apply to those based on deep learning. Nevertheless, results have shown classification accuracies which are not dramatically different from those using other classification techniques. In addition, deep learning techniques generally require very large datasets in terms of available features and training data, and very large computational resources. With that said, decision-level fusion methods including the one proposed in this study can be used together with deep learning networks. 

The remainder of the paper is structured as follows: in Section 2, the study area and images used are described; the methodology including data processing, feature extraction and selection, and the developed ensemble classifier is documented in Section 3; results and a discussion are presented in Section 4 and Section 5, respectively; in Section 6, conclusions and future work are provided. 

## 2. Study Area and Images Used

The study area was selected from a location in Northern Alberta due to the availability of the Alberta Biodiversity Monitoring Institute (ABMI) wetland inventory [54]. Figure 1 illustrates the rough approximate area of interest. This wetland inventory comprises five different land cover classes (fens, bogs, marshes, swamps, and upland), identified and mapped out using photo interpreted data [54]. The ABMI wetland inventory is parsed out in individual study areas throughout Northern Alberta, Canada, each approximately 21 km^2^ in size. For this study, 10 sites, as shown in Figure 1, were selected because of the domination by wetland cover types. Collection and analysis for the photo data were completed in 2016 [54].

The land-cover classes identified in the ABMI wetland inventory (bogs, fens, marshes swamps, and upland) and their detailed descriptions can be found in [47]. Below, the characteristics of these cover types more relevant to remote sensing data interpretation are summarized. Bogs are hydrologically isolated peatlands, receiving water from precipitation only. They are stagnant, with low nutrient availability, and support low biological diversity. Bogs typically have a low water table, appearing dry at the surface.

Fens are also peatlands, but hydrologically connected. Fens can be nutrient-poor or -rich, depending on nutrient input from water sources. Fens often have high water tables and connect wetland systems over great distances. Marshes are mineral wetlands. They exhibit a variable water table, which can vary throughout the season. Marshes receive water from a combination of ground water, runoff, and precipitation, as well as through connecting streams. They are periodically dry, with nutrient-rich soil, promoting the growth of a diverse range of emergent, grass-like vegetation. Swamps are considered mineral wetlands, although they may also exist as peatlands in some cases, with woody plant cover that comprises more than 25% of the total area. Swamps receive water from a combination of ground water, runoff, and precipitation. Water movement ranges from stagnant to dynamic. Swamps typically represent transition zones between other wetlands and non-wetland areas, known as uplands, and support high biological diversity.

The upland class is a broad non-wetland class created by the ABMI in order to encompass non-wetland land covers such as grassy areas, cleared areas, and sparse and dense forests of various species. This includes upland deciduous, mixed-wood, and coniferous stands (age classes combined), grassy areas, and shrub areas [54]. 

For individual study areas shown in Figure 1, ground-truth data were provided in the ABMI dataset. The number of labeled pixels for each study area was determined from the size of land-cover plots identified by ariel imagery and ground survey data as per the ABMI wetland maps. As shown by the example of areas identified as swamp in Figure 2, the labeled pixels were clustered in areas. In the selection of training and validation data, the groupings of pixels were maintained. On average, approximately 64% of the identified pixels were used for training, with the remainder used for validation. These pixel and land-cover assignments are summarized in Table 1. 

Landsat-8, Sentinel-2, and Sentinel-1 imagery represented the primary image sources used in this study. Attempts were made to acquire imagery close to 2016–2017 to match the collection dates of the aerial imagery used to create the ABMI dataset. However, additional images from other dates were also collected in order to create a more robust dataset. It should also be noted that Sentinel-1 imagery coverage of the study area was not available until 2017. The Landsat-8 series of sensors collect multispectral optical imagery with a spatial resolution of 30 m by 30 m across all spectral bands, including bands centered on the thermal spectrum [55]. All Landsat imagery used was Level 1G, which is both radiometrically and geometrically corrected. 

The Sentinel-2 imagery used was the Level 2A bottom-of-atmosphere reflectance in the cartographic geometry imagery product. These images have a resolution of 10 m by 10 m and contain four bands. These bands are centered on 492.4 nm, 559.8 nm, 664.6 nm, and 832.8 nm—blue, green, red, and near-infrared (NIR) respectively [56]. Sentinel-2 imagery was chosen due to its availability, higher resolution compared to Landsat-8, and spectral bands which are useful in characterizing both vegetation and water.

Sentinel-1 imagery (C-band) had a resolution of 5 m by 20 m [57] and two channels in VV and VH. Sentinel-1 imagery was resampled to 10 m by 10 m in order to facilitate ease of analysis with the other imagery products. 

Lastly, a digital elevation map (DEM) of the study area taken from the Canadian Digital Surface Model [58] at a spatial resolution of 30 m by 30 m with an associated DEM derived slope was used. 

In total, three Landsat-8 images, seven Sentinel-2 images, and four Sentinel-1 images were collected. Table 2 summarizes the dates and types of imagery that were collected for this study. 

## 3. Methodology

In this study, an ensemble classifier using a feature- and decision-level fusion framework was developed. Leveraging prior knowledge and all available data in the study area, three classifiers were first designed to reliably distinguish individual or compound classes among the five cover types (fen, swamp, marsh, bog, and upland), executed in parallel with one another using a RF classifier, and the results from these classifiers were then combined according to the D–S theory. The base of this ensemble classifier was the commonly used (also referred to as the traditional method) feature-based fusion method (Classifier #1) for the classification of the five individual classes (fen, swamp, marsh, bog, and upland). As discussed later, with the traditional method, some features known to have high discriminant powers in separating some broad classes (such as wetland and upland) are often not selected using automatic feature selection methods. This may lead to some confusion between wetland and upland classes due to the absence of these features. To overcome this problem, two additional classifiers (Classifiers #2 and #3) were designed to classify compound cover types. For Classifier #2, two broad cover types were classified—wetland (fen, swamp, marsh, bog) vs. dry land covers (upland). For Classifier #3 the focus was on separating more structured land covers (swamp and upland) to less structured land covers (fen, bog, and marsh). Due to the uncertainty expected from any classification method, the D–S theory was employed to combine the results from these classifiers. 

Figure 3 outlines the overall workflow for our approach, and the details are described below.

### 3.1. Features and Their Derivation

Furthermore, 184 candidate features were derived and are summarized in Table 3. The calculations and analyses performed to produce these features are described below.

For this study, 11 different types of remotely sensed features were used. They were vegetation indices, surface albedo, and textual measures derived from multi-spectral imagery, surface temperature from the thermal bands of multi-spectral imagery, backscatter coefficients and derived features from SAR imagery, and digital elevation models (DEMs) and features derived from DEMs. These features were selected in order to characterize vegetative activity, water content, radiometric absorption, horizontal structure and roughness, water content of surface objects, and topography. It is worth mentioning that textual features derived from Sentinel-2 imagery were selected due to their success in the classification of land covers in the popular literature [40,59,60,61] and from our own observations. This is further expanded upon in Section 4 and Section 5. Surface temperature, from our past study [27], was shown to be useful in classifying wetland types.

Specifically, the vegetation indices used included the normalized difference vegetation index (NDVI), the enhanced vegetation index (EVI), and the near-infrared reflectance vegetation index (NIRv). NDVI is a popular and standard vegetation index sensitive to leaf area index, coverage, pigment content of vegetation canopies, and vegetative photoactivity [62,63]. EVI is defined as
(1)$$\mathrm{EVI}=\frac{2.5\times \left(R_{B8}-R_{B4}\right)}{R_{B8}+6\times R_{B4}-7.5\times R_{B2}+1},$$
where $R_{B2}$, $R_{B4}$, and $R_{B8}$ are the reflectance at spectral bands 2, 4, and 8, from Sentinel-2 imagery, respectively. EVI was not calculated using Landsat-8 imagery due to early tests which found that EVI using Sentinel-2 imagery was of a much greater significance during classification. EVI has been shown to be effective in characterizing vegetation features such as leaf area index, temporal changes in vegetative activity and resolving vegetation differences from areas which have complex background surface reflectance [64,65,66]. NIRv, a near-infrared reflectance vegetation index, is defined as
(2)$$\mathrm{NIRv}=\frac{R_{B8}-R_{B4}}{R_{B8}+R_{B4}}\times R_{B8}.$$

Its success in characterizing vegetation in a mixed pixel environment and low leaf areas has been reported in the literature [67]. Again, NIRv was calculated for Sentinel-2 imagery only due to its larger significance when compared to NIRv calculated with Landsat-8 imagery. NDWI works on a similar principle to NDVI but is designed to be sensitive to water content rather than to photosynthetic activity. NDWI is defined as
(3)$$\mathrm{NDWI}=\frac{R_{B5}-R_{B6}}{R_{B5}+R_{B6}},$$
where $R_{B5}$ and $R_{B6}$ are the reflectance in the green and mid-infrared band (MIR), respectively, from Landsat-8 imagery. NDWI was only calculated for Landsat-8 imagery because early tests showed that, for Sentinel-2 imagery in our study area, NDVI was a much more significant feature compared to NDWI. The authors of [68] asserted that NDWI is more sensitive to changes in liquid water content of vegetation canopies vs. NDVI. They [68] also argued that the effect of atmospheric aerosol scatter effects in the MIR region are weak; thus, NDWI is less sensitive to atmospheric optical depth compared with NDVI. Due in part to its success in the popular scientific literature, NDWI is a standard layer product for the Moderate Resolution Imaging Spectroradiometer (MODIS) sensor [69].

Surface albedo is a measure of reflectivity from a surface, ranging from 0 (full absorption) to 1 (complete reflectance). A standard approach in determining the surface albedo using Landsat imagery is through a numerically determined relationship described by Liang et al. [70,71]. Liang described albedo α using Landsat-5 TM imagery through the following equation:(4)$$\alpha =0.356\alpha_{1}+0.130\alpha_{3}+0.373\alpha_{4}+0.085\alpha_{5}+0.072\alpha_{7}-0.0018,$$
where the subscript on each α represents a band number in a Landsat-5 TM image. For Landsat-8 imagery the band subscripts were $\alpha_{2}$,${\;\alpha }_{4}$,$\;\alpha_{5}$,${\;\alpha }_{6}$,${\;\alpha }_{7}$ vs. $\alpha_{1}$,${\;\alpha }_{3}$,${\;\alpha }_{4}$,${\;\alpha }_{5}$,${\;\alpha }_{7}$ for Landsat-5. 

Surface temperature was calculated for individual pixels from Landsat-8 imagery using the standard methodology from the Landsat-8 (L8) Data Users Handbook [55]. 

The textural features were derived from Sentinel-2 imagery, due to its relatively higher spatial resolution in comparison with that of Landsat 8. The three texture features (mean, variance, and entropy) were calculated within a window size of 4 × 4 pixels for the four Sentinel-2 imagery bands, using the standard software suites in ENVI 5.6 [65], and they are defined in Equations (5)–(7). This window size was determined empirically.
(5)$$\mathrm{Mean}=\overset{N_{g}-1}{\underset{i=0}{\sum }}\mathrm{iP}\left(i\right),$$
where Ng is the number of distinct grey levels in the quantized image, and $P\left(i\right)$ is the probability of the occurrence of each gray level [72].
(6)$$\mathrm{Variance}=\overset{N_{g}-1}{\underset{i=0}{\sum }}{\left(i-M\right)}^{2}P\left(i\right),$$
where M is the mean as defined in Equation (5) [72].
(7)$$\mathrm{Entropy}=-\overset{N_{g}-1}{\underset{i=0}{\sum }}P\left(i\right)\times \mathrm{lnP}\left(i\right).$$

In these equations, Ng is the number of distinct gray levels in the quantized image, $P\left(i\right)$ is the probability of the occurrence of each grey level, and M is the mean as defined in Equation (5) [72]. In this study, Ng was determined automatically by ENVI through the available quantization range of the imagery.

The backscatter coefficients in VV and VH, denoted as $\sigma_{\mathrm{vv}}$ and $\sigma_{\mathrm{vH}},$ respectively, were obtained from the calibrated Level 1 Single Look Complex (SLC) product of Sentinel-1 [57]. In order to reduce noise, the enhanced frost speckle filter from PCI Geomatica with a 5 × 5 pixel window was used to filter all Sentinel-1 imagery. The window size was chosen on the basis of empirical analysis. After processing, the Sentinel-1 imagery was georeferenced to the Sentinel-2 imagery.

From the Sentinel-1 imagery, we also produced an adaption of the quad-polarization of the SAR vegetation index (RVI) proposed by Periasamy [73], i.e., the dual-polarization SAR vegetation index (DPSVI), defined as
(8)$$\mathrm{DPSVI}=\frac{\sigma_{\mathrm{vv}}+\sigma_{\mathrm{vH}}}{\sigma_{\mathrm{vv}}}.$$

This index has been found to be a significant feature in separating different types of crops and from separating land covers of high vegetation water content from land covers better characterized by dry biomass.

Additionally, the DEM, DEM-derived slope, and valley bottom flatness (VBF) were used. Slope was calculated from the DEM using the ENVI 5.6 topographic modeling function with a 3 × 3 window. The DEM and DEM-derived slope were selected to determine the role geographic features play in distinguishing wetland classes. For instance, it is known that some species of fens prefer to grow on slopes. VBF was calculated using the open-source GIS software suite System for Automatic Geoscientific Analysis (SAGA) using the processed DEM data as previously described. VBF measures the degree of valley bottom flatness at multiple scales [74]. Large flat valleys are typical of landscapes for wetlands, once open water has been masked from the data. Experiments were conducted while varying slope thresholds, where it was found that a slope threshold of 17 produced the most significant VBF feature. VBF has been found to be a very significant feature in the classification of wetlands from the ABMI dataset, as reported by the Alberta Biodiversity Monitoring Institute [54].

As a final note regarding the imagery and features used in this study, in order to preserve the information from the higher-resolution Sentinel-2 imagery, all images were resampled to 10 m by 10 m resolution when layers were stacked together.

### 3.2. Feature Selection

As mentioned earlier, three classifiers were designed in this study, and two feature selection methods were employed. For Classifier #1 (see details in the next section) where all cover types were identified, the built-in feature selection mechanism in the RF classification was used. This was to fully utilize the abovementioned extracted features and maximize their discriminant power in the classification. For Classifiers #2 and #3, where broad cover types (compounds of cover types) were considered, the feature selection was conducted on the basis of prior knowledge and experimentation. This was applied to maintain the independency in the features used for these three classifiers to avoid any bias in the fusion process. Furthermore, it was also noted in a previous study [27] that a subset of an analyzed and ranked set of features could be outperformed, in a classification setting, by a set of features selected through a holistic approach. This is further expanded upon in Section 5.

The RF importance value is determined through an iterative exploration of the dataset [75]. It is computed by summing changes in the percentage increase in mean squared error (MSE) due to splits on every predictor and dividing the sum by the number of branch nodes for that tree, averaged over all trees. These calculations are performed on all input features, with larger values implying that a feature is more significant. Additionally, it was observed from our previous study [27] that there was a plateau in classification accuracy once a specific number of image features was reached. With the increase in the number of features in a given classification, it was likely that the redundance among those features was increased, implying that there is a ceiling to the classification accuracy for a given dataset. Furthermore, any noise and confliction among a large number of features might negatively affect the classification accuracy. Keeping the aforementioned in mind, when selecting sets of features, we were cognizant of identifying the appropriate number of features in order to avoid redundancy and noise. This is further expanded upon in Section 5.

### 3.3. The Ensemble Classification Method Based on the D–S Theory

Our previous investigation [28] showed that, in the context of wetland classification, in a hierarchical framework, certain features can separate and classify a group of wetland types more effectively and more reliably when compared to a group focused on distinguishing individual types. One disadvantage of a hierarchical framework lies in the fact that the misclassification in the higher hierarchy is propagated to the subsequent levels of classification. To address this issue, three classifiers with different propositions were designed and carried out first, and their results were then combined according to the D–S theory. In this way, the uncertainty associated with each classifier was considered. 

In Classifier #1, individual wetland cover types were considered. The classification propositions were fen, bog, marsh, swamp, and upland. This classifier type is commonly used; thus, it was taken as the baseline method for comparison. For Classifier #2, two broad cover types were classified—wetland vs. dry land covers. This classifier would utilize features which excel at identifying moisture, and flat structural features in pixels such as water indices, SAR backscatter coefficients, and DEM and its derivatives. For Classifier #3, the focus was on separating more structured land covers (swamp and upland) from less structured land covers (fen, bog, and marsh). For Classifier #3 the use of SAR features and textural features was leveraged given their performance advantages in those areas. 

As mentioned in the previous section, for Classifier #1, a suitable set of features were selected using the RF feature selection method. For Classifiers #2 and #3, feature selection was conducted on the basis of prior knowledge in the separation of the two broad classes. 

The RF classifier is an ensemble, supervised, machine learning algorithm. It operates by constructing multitudes of decision trees with the ultimate class of a given input determined by the majority vote from those decision trees [75,76,77]. With RF, diversification of the decision trees is accomplished by developing those trees from various subsets created through bagging or bootstrap aggregating from the training data [76]. RF lends itself well to parallelization and computational streamlining for investigating the nuances of large datasets. This has led RF to become one of the most successful and widely implemented machine learning algorithms to date [75,76]. RF generally requires two main input parameters: the number of trees to grow and the depth or complexity of those trees (*p*-value). More trees generally result in higher classification accuracies but at greater computational costs. However, at some point, increasing the number of trees no longer increases classification accuracy. Similarly, choosing a tree depth that is too shallow tends to produce trees that underfit, whereas choosing trees that are too deep will overfit the data. A total of 150 trees were used as determined through experimentation. A *p*-value of 0.05 was determined using the curvature test, which is utilized with the RF classifier to determine when to terminate a split in a decision tree. The aforementioned techniques used to determine RF input parameters is considered to be a standard approach [75,76,77,78]. 

For the results generated from RF classification, not only was the class assignment for each pixel generated, but also the posterior probability, which was treated as the mass function within the framework of D–S theory. 

The D–S theory is a general framework for reasoning with uncertainty. It allows the user to combine evidence from different sources and arrive at a degree of belief (a mass function) that takes into account all of the available, independent, sources of evidence. Given that the RF classifier works on a majority voting principle, one product it produces is a confidence value for each of the possible outcomes based on the percentage of votes. We utilized this confidence value as a measure of belief in that outcome in the context of the D–S framework. When executing computations with the D–S rule, for each of our classifiers, we treat it as a proposition in the D–S framework. The D–S rule states that
(9)$$\left(A\right)=\frac{\sum_{B_{1}\cap ...\cap B_{n}=A}m_{1}\left(B_{1}\right)\ldots m_{n}\left(B_{n}\right)}{1-K},\;K=\underset{B_{1}\cap ...\cap B_{n}=\emptyset }{\sum }\overset{n}{\underset{i=1}{\prod }}m_{i}\left(B_{i}\right),$$
where $m\left(A\right)$ is the mass function of a proposition A after considering n pieces of evidence (in our case, the different classifiers), $m_{i}\left(B_{i}\right)$ is the mass function in the proposition $B_{i}$ supported by the *i-*th piece of evidence, and K is known as the total conflict factor [79]. As shown in Figure 3, the three classifiers mentioned earlier were first computed using the RF classification method; their results were then combined using the Dempster rule of combination under the D–S framework. The final classification was produced by assigning a given pixel to the class with the maximum mass function. As part of the analysis of the final classification result, comparisons were made to examine changes in land-cover assignment, and to see how the number of high-confidence misclassified pixels changed from the standard classifier (Classifier #1). 

## 4. Results

Table 4 summarizes the features selected for Classifier #1 resulting from the feature selection method described in Section 3.2, as well as those for Classifiers #2 and #3. On the basis of these features, the classification accuracies were 87.5%, 88.3%, and 89.5% for Classifiers #1, #2, and #3, respectively. 

When examining Table 4, we can note that, for Classifier #1, there were a broad mix of features from different sources and image types. For Classifiers #2 and #3 (broad class separations) a more limited and specific set of features were used. It can be further noticed that the features employed in Classifiers #2 and #3 were mostly excluded from the feature sets automatically selected for Classifier #1. This demonstrated that important features that could be used to distinguish compound cover types would not be employed at all using the traditional feature-level fusion method. Additionally, having different groups of features utilized in these classifiers indicated independence in their classification results, which is important within the framework of the D–S theory.

Classification maps generated by the proposed ensemble method for two selected tested areas dominated by upland and wetland are shown in Figure 4 and Figure 5, respectively, together with the true-color composite of Sentinel-2 imagery and ground-truth maps. The misclassification pixels by the traditional method and those that were corrected by the proposed ensemble method are highlighted in Figure 4D and Figure 5D. Observing Figure 4A,B and Figure 5A,B, it can be observed that the classification results generated using the ensemble method were consistent with the ground-truth maps and visual observations. It can be further noted that the misclassification using the traditional method (Classifier #1) was clustered in the upland area (Figure 4C) and fen area (Figure 5C), both in locations with high spatial variation, and the majority of the misclassification pixels were corrected by the addition of Classifiers #2 and #3 (the ensemble method).

In addition to the visual assessment of the classification results, quantitative analysis was carried out and the confusion matrices for the traditional method and the Ensemble Classifier are shown in Table 5 and Table 6, respectively. In this section, we will focus on observations from these results, while detailed discussion will be provided in the discussion section. 

In general, the ensemble classifier incorporating the three classifiers together based on the D–S theory resulted in an increase in the classification accuracy from 87.5% (the traditional method, Classifier #1) to 93.5%. Upon closer examination of the results using the traditional method (Table 5), it can be noted that the producer accuracy was high and fairly uniform across all land covers. However, the user accuracy was lower; in particular, it was the lowest at 14% for swamp. When examining the results using the proposed method (Table 6), it can be noted that the user accuracy for swamps was increased by ~0.18 to 0.32. In addition, it can be observed that the upland land cover was misclassified the most in terms of the number of raw pixels and as a percentage of pixels misclassified. This might be due to the broad nature of the upland class. 

Checking the misclassified pixels, it can be noted that, for some, the support (mass function) for the “wrong” cover type was very strong (over 0.85), indicating a high confidence for the class assignment. However, it was observed that there was a reduction in the number of high-confidence misclassified pixels from 26222 to 23,588—a reduction of ~10% using the proposed method (Table 7). These results show that the addition of two classifiers with compound classes through the ensemble classifier provided value in increasing the accuracy and decreasing the number of the incorrectly classified pixels with high confidence. 

To further examine the improvement in individual land-cover classification provided by the ensemble classifier, tables to show changes in the pixel assignments for each cover type were generated (Table 8 and Table 9).

It can be noted from these tables that the majority of misclassified pixels, across all classes, which were reclassified by the ensemble classifier, were moved to the upland class. Of additional note, a large number of pixels originally assigned to swamps were moved to other classes, including the upland class. This movement in the assignment of pixels would also explain the large increase in user accuracy for swamps by the proposed ensemble classifier. Among these misclassified pixels with their assignment changes, some of them were classified correctly using the proposed method, while some were still misclassified, and the correct class had the second strongest support from the evidence. However, for some in the latter group, the classification uncertainty was high. That is, for these pixels, the largest mass function was not significantly different from the second largest one (difference between 0.05–0.10), leading to large uncertainty class assignment. These pixels were also summarized, as shown in Table 9. 

## 5. Discussion

### 5.1. On Feature Selection and Selected Features for Classification

In this study, the selection of features for Classifier #1 followed a standard data-driven machine learning methodology, which is commonly used. The features for Classifiers #2 and #3 were manually selected, following a holistic approach, similar to that presented in a previous paper of ours [27]. From a holistic standpoint, we selected families of features which, by design, were best suited for class separation sought for each classifier, while ensuring the independence of these classifiers required by the D–S theory. The design of Classifiers #2 and #3 in terms of class propositions was to fully utilize the available datasets. It was observed that, for Classifier #1, most features selected were from optical imagery. For instance, backscatter coefficients and related indices from SAR imagery and water indices from optical imagery were known and, thus, identified for Classifier #2 (separating wetlands from upland covers), while backscatter coefficients from SAR imagery and textural features from optical imagery were identified for Classifier #3 (separating structured from less structured land covers). Through feature analysis and experimentation, we were able to determine a set of features which maximized the classification accuracy for those classifiers. As an interesting note, in the previous study [27], we reported that there were many instances where a set of holistically determined features actually produced more accurate classification results when compared to sets of features selected through quantitative analysis. In this study, we also observed the same phenomenon when determining feature inputs for Classifiers #2 and #3. These results may call for an integrated knowledge-based and data-driven feature selection method. These results also confirmed our belief (briefly mentioned in Section 1) that simple feature-level fusion for classification using multi-source remotely sensed data might underutilize some features. 

From an imagery standpoint, it was noted that there was no clear correlation between the collection dates of the imagery and their significance. Intuitively, imagery closer to 2016 (the collection date of the aerial imagery used to create the ABMI plots) should be of greater significance but this was generally not the case. Landsat imagery from 2015 and 2016 was more significant when compared to the collection from 2020, while, for both Sentinel-1 and -2, there was no clear correlation. This might indicate that features of these cover types exhibited in Sentinel-2 SAR imagery were not highly dynamic. It was also suspected that factors such as atmospheric attenuation and inter-year variations in water levels were due in part in driving these differences. 

By exploring images, it was also noted that classification accuracies from inputs strictly drawn from Landsat-8 images produced accuracies which were higher when compared to classification experiments where inputs were strictly drawn from Sentinel-2 images. This was counter-intuitive. It would be expected that inputs with higher resolution would result in higher classification accuracies. However, upon further analysis, it was noted that the land covers considered in this study were broader when compared to other land cover maps which have more narrow class definitions [80]. In previous studies, by virtue of data availability, land covers such as fens and bogs were parsed further into treed and non-treed versions of those land covers. For those datasets, the higher-resolution imagery might have provided the expected accuracy increases; however, with this ABMI dataset with broader classes, it is suspected that the high spatial resolution of the Sentinel-2 images might have introduced more variability among cover types, which made the classification it more difficult. Lastly, during these experiments, it was noted that the classification accuracy when using only individual datasets was some 5–8% lower when compared to classification accuracies from multi-source remotely sensed data, which is consistent with the literature. 

### 5.2. On Misclassified Pixels

The core of this study was the development of the ensemble classifier in an effort to increase classification accuracies while also reducing the number of the incorrectly classified pixels with high confidence. The prevalence of misclassified pixels of high confidence (>85% certainty in assignment) and misclassified pixels which had the correct land cover class as the second highest ranked land cover was noted in this study with Classifier #1 (the traditional method). As shown by the results, these issues were overcome by adding two classifiers in the proposed ensemble classifier to a certain extent. Examining the mis-classified pixels using the proposed method, it was noticed that the three classifiers were not always in agreement with one another, as shown in Table 10. It was further noticed that the misclassified pixels with high confidence were located at the transition zones between cover types, as shown in Figure 6. This intuitively and physically makes sense since the transition from one wetland cover type to another is fuzzy in nature [81]. 

In addition to the transition zones where these classifiers tended to conflict with each other, pixels with disagreement among classifiers were also within in the areas with high variability according to a visual examination, as shown in Figure 7. This would drive variations in features, which in turn could then contribute to the variability in the outcomes of the different classification propositions. Additional information may be needed to further solve this confliction. 

The misclassification involving upland may also be due to the fact that the upland class was very broad and encompassed a great deal of different land-cover types, which led to large variations in the selected features for it. As an example, Classifier #3 was used to classify structured and nonstructured cover type. Upland was included in the structured class, considering the domination of trees and shrubs in this class. However, there were also nonstructured cover types in this class. To mitigate this, we attempted to split the upland class to two subclasses during the decision-level fusion process according to the D–S theory. However, there was no real improvement in the results (not shown). The best strategy was to separate upland to different categories, which was not attempted due to the lack of training samples for detailed upland cover types. 

### 5.3. On the Proposed Ensemble Classifier 

The overall classification accuracy of the proposed method was 0.93. When compared to other studies, it was noted that this accuracy was greater or comparable with those obtained for land-cover classification using multi-source remotely sensed data [34,82,83,84]. In addition, the proposed method was less complex than some of these studies. It should be stressed that we could not find classification studies of our study area which used a decision- or feature-level fusion framework for direct comparison in the literature; furthermore, all of the comparable studies we found used different datasets or combinations thereof, for both the land-cover maps and the remotely sensed imagery used. However, it can be noted that the ABMI conducted its own classification studies of its own dataset using Landsat-5 and Landsat-8 imagery, with an RF classifier; the classification accuracies were around 0.8–0.85 [54]. 

A direct comparison was carried out in this study with the traditional classification method based on feature-level fusion using multi-source remotely sensed data (Classifier #1). Results were presented and discussed in the previous sections. The improvement of the proposed method over the traditional method relied on its effective utilization of available datasets and features. As previously mentioned in Section 1, features that can be used to separate certain cover types might be excluded by considering all cover types together, such as the features derived from SAR imagery and DEM. The inclusion of these features otherwise excluded in Classifiers #2 and #3 led to an increase in in user accuracy of the swamp class by ~18%. It was also noted that, in Classifier #1, the impact of the SAR imagery was lower when compared to it being utilized in a classifier focused on broader class separations. When combined in the ensemble classifier, the value of this imagery was better utilized. 

In total, the proposed ensemble classifier provided a framework to effectively utilize the best available data in order to support wetland classification. In this study, while an RF classifier was employed, other classifiers could be utilized. We experimented with support vector machine (SVM) and naïve Bayes classifiers, where we found that the overall accuracies were generally lower (by ~5–8%), but the computation times were greatly reduced, compared to similar RF tests, in some cases by over 80%. The additional classification accuracy gained by using RF was obtained at a considerable computational cost.

The proposed ensemble classifier combined the strengths of various types of remotely sensed data in the differentiation of wetland cover types. This same principle could be applied to the classification of other cover types. The three classifiers were designed parallelly and independently, even though the same classifier (RF) was used. The idea of designing two classifiers to classify broader cover types was inspired by the hierarchical classification methods including our own work. As mentioned earlier, with hierarchical classification methods, the errors/uncertainties in the higher hierarchies are often not considered in the lower ones; thus, error propagation is the biggest problem. With the proposed method in this study, the uncertainty associated with these classifiers was explicitly considered under the framework of the D–S theory and, thus, solved the error propagation problem in hierarchical classification. In the same vein, this study expands the literature on the utilization of the D–S theory. Even though the D–S theory is powerful conceptually, its application is not trivial, especially in the determination of the mass functions, including the selection of propositions of non-zero mass functions. As mentioned in Section 1, in most studies based on the D–S theory, identical sets of classes were often employed for different classifiers [30,34,52,53]. Different prepositions were considered for the three classifiers in this study, and they were selected according to prior knowledge of the wetland cover types and remotely sensed data. Not only did this result in higher classification accuracies but it also provides us a framework for future work where we can more easily explore subclasses, class overlaps, and unknown classes.

The usage of prior knowledge in the designing of Classifiers #2 and #3 was also one of the disadvantages of the proposed approach. In this study, the categories and features were selected manually. This may not be practical for studies dealing with a large number of cover types. Ideally, a knowledge-based automatic approach would be preferable. This will be pursued in future work. 

Lastly, the ensemble classifier in its current form has not been successful in dealing with cover types with great diversity such as the upland class. Even though it is ideal to separate such cover types into different several classes during the training process, it might not be realistic due to the difficulty in the selection of training samples. Our initial experiments where we tried to separate two subclasses during the decision-level fusion process did not show accuracy increases or effective or consistent class separation. Future versions of this classifier will have to address this.

## 6. Conclusions

An ensemble classification methodology combining three classifiers based on the D–S theory was developed and tested on a study area in Northern Alberta. Classifier #1 was a traditional feature-level fusion method for classification using multi-source remotely sensed data where all land-cover classes were classified together. The other two classifiers were focused on compound cover types. With Classifier #2, wetland cover types (fen, bog, marsh, and swamp) and dry land covers (upland) were considered, whereas, with Classifier #3, the focus was on the separation of less structured land covers (fen, bog, and marsh) and more structured ones (swamp and upland). Features used for classification were determined using the analysis of RF feature significance for Classifier #1 and through a more holistic approach for Classifiers #2 and #3. Use of a holistic approach for feature selection was not traditional; however, on the basis of prior knowledge and experimentation, we were able to select a set of features for Classifiers #2 and #3 which produced high accuracy when compared to a strict feature significance analysis approach. This also mimicked the results observed in past studies [27]. Once each classifier was computed, those results were combined using the Dempster’s combination rule. Results showed that the proposed ensemble classifier increased the classification accuracy from 0.88 to 0.93, compared with the traditional classification method (Classifier #1). Additionally, it was noted that there was a reduction of ~10% in the number of the misclassified pixels with high confidence, which provides additional assurance in the quality of the classification results, something which is generally not explored in this style of research.

The proposed approach provided a framework to intelligently utilize available remotely sensed data for wetland classification, which could be employed for other cover type classification. Incorporating data-driven machine learning and knowledge-based holistic methods, different propositions were designed; thus, different features were selected for these three classifiers to maximize their discriminant powers in the classification of these wetland cover types (individually or in combination). As detailed in the discussion, this made this framework unique compared with most studies based on D–S theory reported in the literature. In addition, compared with hierarchical classification methods, the proposed ensemble classifier’s advantages were enhanced by selecting different features to classify different classes, while its weaknesses were addressed by explicitly taking into account the uncertainties of different classifiers. 

Even though the holistic knowledge-based method was successful in the design of Classifiers #2 and #3, prior knowledge could be utilized in a more explicit and automatic fashion, enabling the proposed method to be employed as a general framework in wider applications. This will be endeavored moving forward. With the current approach, advanced features derived from the available datasets will be further explored, and more classifiers will be added. Additional testing will be also carried out by expanding the study area to the remaining parts of the ABMI wetland inventory. Other data sources, such as RADARSAT-2 and LiDAR images, will be considered. 

## Figures and Tables

**Figure 1 sensors-22-08942-f001:**
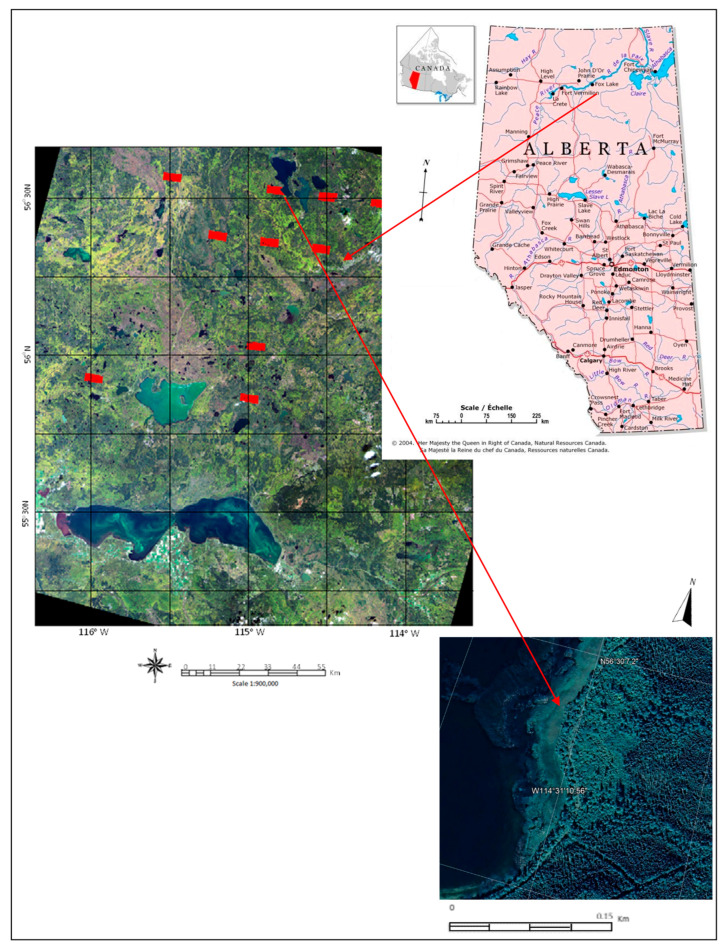
The study area from a geographic perspective together with a Landsat-8 True-Color image (RGB Bands 4, 3, 2) and aerial imagery. Individual study areas are highlighted as red polygons, drawn from the ABMI wetland inventory dataset on the Landsat-8 image.

**Figure 2 sensors-22-08942-f002:**
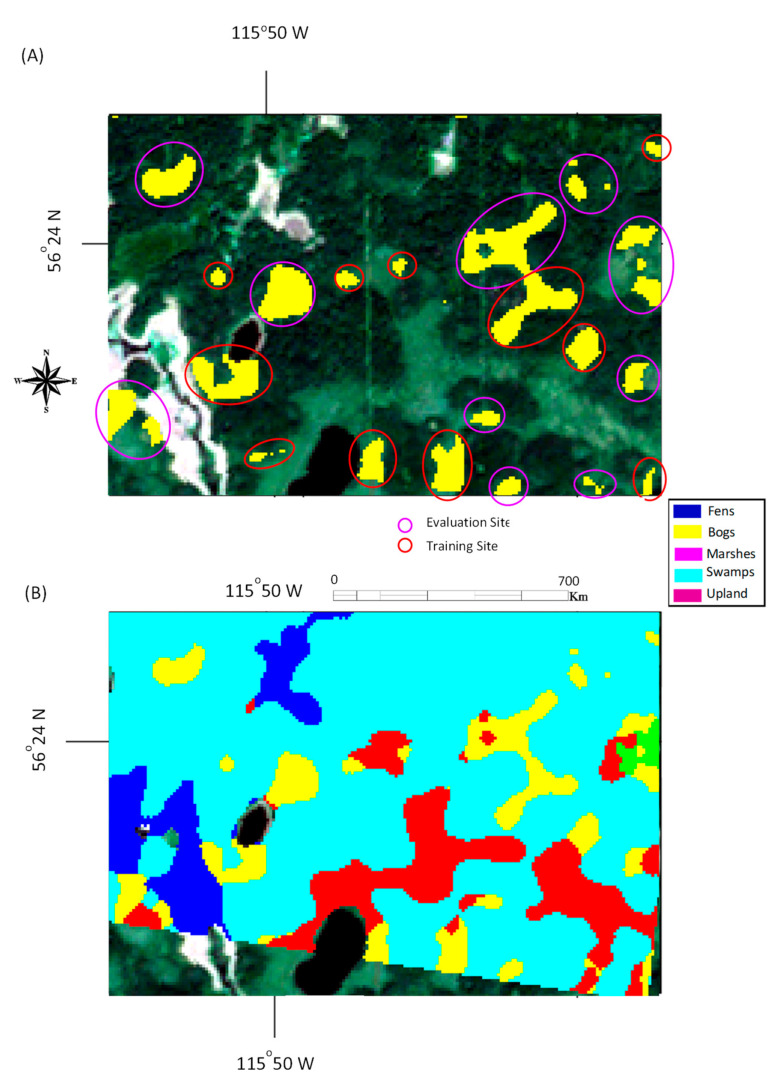
Example of a study area highlighting the individual evaluation and training sets for swamps (**A**) and its corresponding cover type map (ground truth) (**B**).

**Figure 3 sensors-22-08942-f003:**
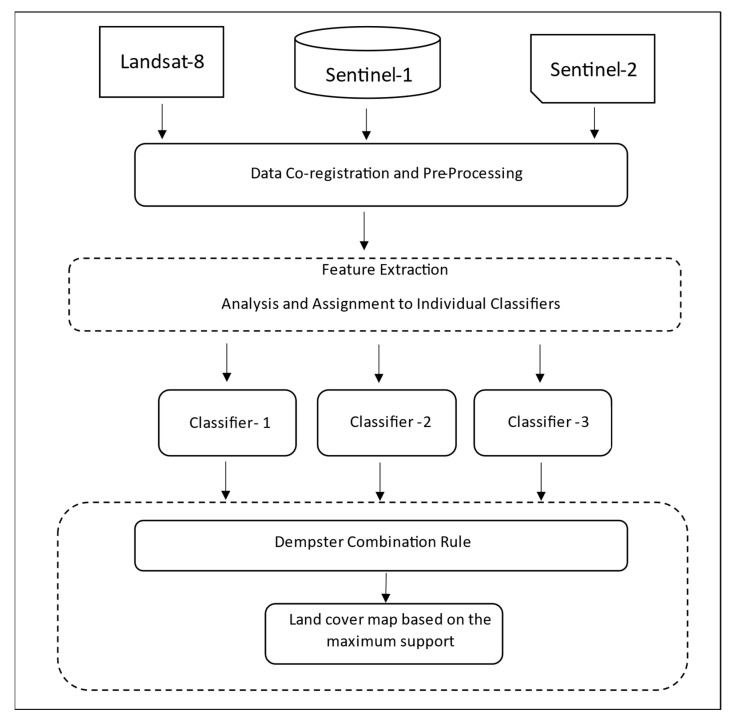
Workflow of the proposed ensemble classifier combining the results of three different classifiers based on D–S theory.

**Figure 4 sensors-22-08942-f004:**
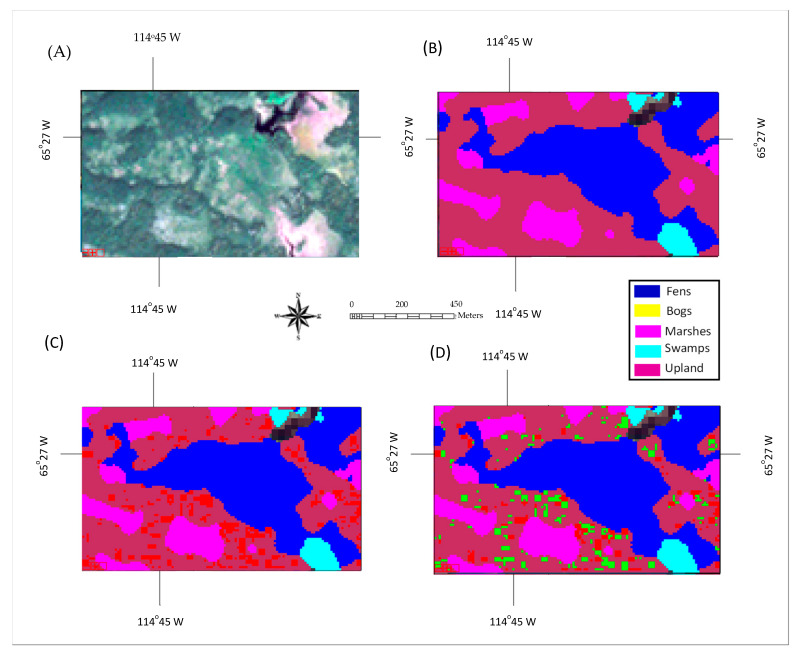
Classification result of a test area dominated by upland: (**A**) true-color composite of a Sentinel-2 image; (**B**) ground-truth-based classification map; (**C**) classification map using the proposed method. Misclassified pixels highlighted in red; (**D**) Classification map using Classifier #1, where the misclassified pixels, which were corrected using the proposed method, are highlighted in green.

**Figure 5 sensors-22-08942-f005:**
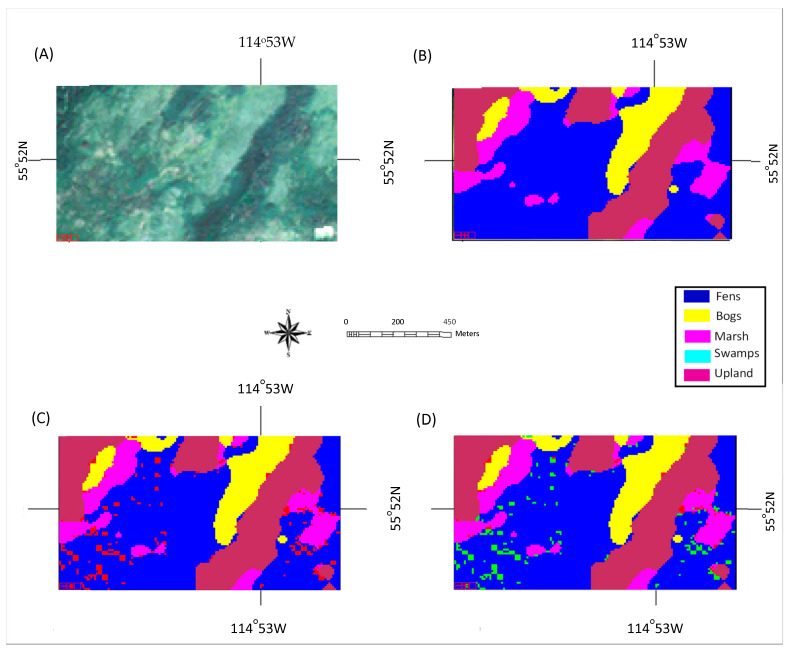
Classification result of a test area dominated by wetland: (**A**) true-color composite of a Sentinel-2 image; (**B**) ground-truth-based classification map; (**C**) classification map using Classifier #1 where misclassified pixels are highlighted in red; (**D**) classification map using Classifier #1, where the misclassified pixels, which were corrected using the proposed method, are highlighted in green.

**Figure 6 sensors-22-08942-f006:**
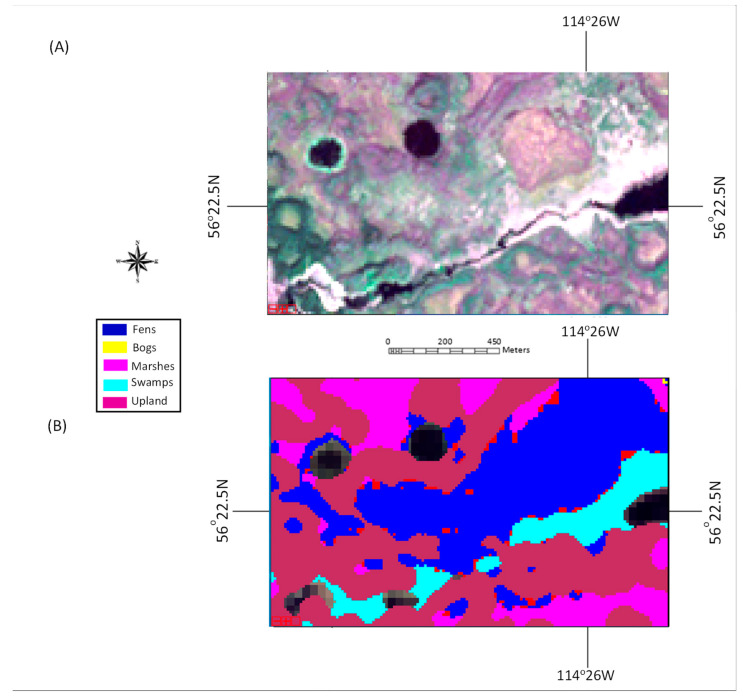
A Test area containing the misclassified pixels with high confidence: (**A**) true-color Sentinel-2 image; (**B**) classification map using the proposed method with the misclassified pixels highlighted in red.

**Figure 7 sensors-22-08942-f007:**
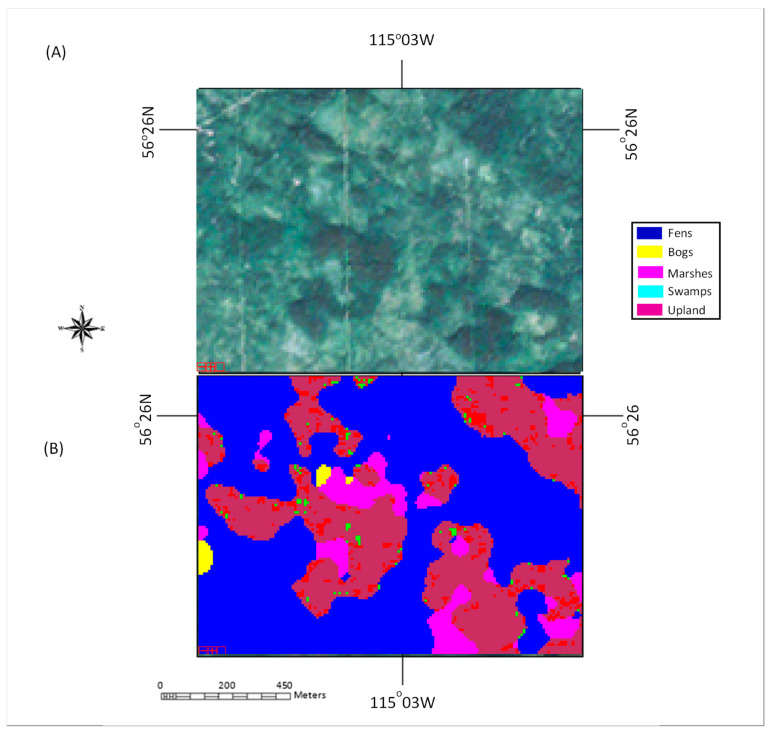
(**A**) True-color image of test area from Sentinel-2 imagery; (**B**) classification map showing pixels where two classifiers disagree or all classifiers disagree, as highlighted in red and green, respectively.

**Table 1 sensors-22-08942-t001:** Land-cover class assignment and the number of pixels contained in the training and validation set.

Class	Number Assigned to Class	Number of Pixels in Training Set	Number of Pixels in Validation Set
Fen	1	288,343	156,102
Bog	2	36,637	14,479
Marsh	3	25,309	23,416
Swamp	4	109,490	91,510
Upland	5	2,314,364	636,441

**Table 2 sensors-22-08942-t002:** Summary of satellite imagery collected for this study.

Image ID	Imagery	Season	Date	Level of Processing	Accessed From
#1	Landsat-8	Summer	27 July 2015	Level 1G	United States Geological Survey (USGS)
#2	Landsat-8	Fall	15 September 2016	Level 1G	USGS
#3	Landsat-8	Fall	10 September 2020	Level 1G	USGS
#4	Sentinel-2	Fall	17 September 2017	Level 2A	European Space Agency (ESA)—Sentinel
#5	Sentinel-2	Summer	11 August 2017	Level 2A	ESA—Sentinel
#6	Sentinel-2	Summer	2 September 2018	Level 2A	ESA—Sentinel
#7	Sentinel-2	Fall	29 September 2020	Level 2A	ESA—Sentinel
#8	Sentinel-2	Summer	28 June 2021	Level 2A	ESA—Sentinel
#9	Sentinel-2	Summer	1 July 2021	Level 2A	ESA—Sentinel
#10	Sentinel-2	Summer	28 July 2021	Level 2A	ESA—Sentinel
#11	Sentinel-1	Summer	12 August 2018	Level 1—SLC	ESA—Sentinel
#12	Sentinel-1	Summer	27 July 2019	Level 1—SLC	ESA—Sentinel
#13	Sentinel-1	Fall	19 September 2020	Level 1—SLC	ESA—Sentinel
#14	Sentinel-1	Summer	9 August 2021	Level 1—SLC	ESA—Sentinel

**Table 3 sensors-22-08942-t003:** Features used during this study and their associated variable index. Reflection is shortened to “Reflect.” and Sentinel is shortened to “Senti.” M, V, and E correspond to the mean, variance, and entropy texture, respectively. The number at the end of each feature name refers to the image ID in Table 2.

Index	Name	Index	Name	Index	Name	Index	Name
1	B1 Reflect. #1	47	B3 Senti. 2 #6	93	B4-M-Senti. 2 #6	139	B2-E-Senti. 2 #4
2	B2 Reflect. #1	48	B4 Senti. 2 #6	94	B1-M-Senti. 2 #7	140	B3-E-Senti. 2 #4
3	B3 Reflect. #1	49	B1 Senti. 2 #7	95	B2-M-Senti. 2 #7	141	B4-E-Senti. 2 #4
4	B4 Reflect. #1	50	B2 Senti. 2 #7	96	B3-M-Senti. 2 #7	142	B1-E-Senti. 2 #5
5	B5 Reflect. #1	51	B3 Senti. 2 #7	97	B4-M-Senti. 2 #7	143	B2-E-Senti. 2 #5
6	B6 Reflect. #1	52	B4 Senti. 2 #7	98	B1-M-Senti. 2 #8	144	B3-E-Senti. 2 #5
7	B7 Reflect. #1	53	B1 Senti. 2 #8	99	B2-M-Senti. 2 #8	145	B4-E-Senti. 2 #5
8	NDVI #1	54	B2 Senti. 2 #8	100	B3-M-Senti. 2 #8	146	B1-E-Senti. 2 #6
9	NDWI #1	55	B3 Senti. 2 #8	101	B4-M-Senti. 2 #8	147	B2-E-Senti. 2 #6
10	Albedo #1	56	B4 Senti. 2 #8	102	B1-M-Senti. 2 #9	148	B3-E-Senti. 2 #6
11	Temp1 #1	57	B1 Senti. 2 #9	103	B2-M-Senti. 2 #9	149	B4-E-Senti. 2 #6
12	Temp2 #1	58	B2 Senti. 2 #9	104	B3-M-Senti. 2 #9	150	B1-E-Senti. 2 #7
13	B1 Reflect. #2	59	B3 Senti. 2 #9	105	B4-M-Senti. 2 #9	151	B2-E-Senti. 2 #7
14	B2 Reflect. #2	60	B4 Senti. 2 #9	106	B1-M-Senti. 2 #10	152	B3-E-Senti. 2 #7
15	B3 Reflect. #2	61	B1 Senti. 2 #10	107	B2-M-Senti. 2 #10	153	B4-E-Senti. 2 #7
16	B4 Reflect. #2	62	B2 Senti. 2 #10	108	B3-M-Senti. 2 #10	154	B1-E-Senti. 2 #8
17	B5 Reflect. #2	63	B3 Senti. 2 #10	109	B4-M-Senti. 2 #10	155	B2-E-Senti. 2 #8
18	B6 Reflect. #2	64	B4 Senti. 2 #10	110	B1-V-Senti. 2 #4	156	B3-E-Senti. 2 #8
19	B7 Reflect. #2	65	Senti. VV-#11	111	B2-V-Senti. 2 #4	157	B4-E-Senti. 2 #8
20	NDVI #2	66	Senti. VH-#11	112	B3-V-Senti. 2 #4	158	B1-E-Senti. 2 #9
21	NDWI #2	67	Senti. VV-#12	113	B4-V-Senti. 2 #4	159	B2-E-Senti. 2 #9
22	Albedo #2	68	Senti. VH-#12	114	B1-V-Senti. 2 #5	160	B3-E-Senti. 2 #9
23	Temp1 #2	69	Senti. VV-#13	115	B2-V-Senti. 2 #5	161	B4-E-Senti. 2 #9
24	Temp2 #2	70	Senti. VH-#13	116	B3-V-Senti. 2 #5	162	B1-E-Senti. 2 #10
25	B1 Reflect. #3	71	Senti. VV-#14	117	B4-V-Senti. 2 #5	163	B2-E-Senti. 2 #10
26	B2 Reflect. #3	72	Senti. VH-#14	118	B1-V-Senti. 2 #6	164	B3-E-Senti. 2 #10
27	B3 Reflect. #3	73	DEM	119	B2-V-Senti. 2 #6	165	B4-E-Senti. 2 #10
28	B4 Reflect. #3	74	Slope	120	B3-V-Senti. 2 #6	166	EVI Senti. 2 #4
29	B5 Reflect. #3	75	NDVI Senti. 2 #4	121	B4-V-Senti. 2 #6	167	NIRv Senti. 2 #4
30	B6 Reflect. #3	76	NDVI Senti. 2 #5	122	B1-V-Senti. 2 #7	168	EVI Senti. 2 #5
31	B7 Reflect. #3	77	NDVI Senti. 2 #6	123	B2-V-Senti. 2 #7	169	NIRv Senti. 2 #5
32	NDVI #3	78	NDVI Senti. 2 #7	124	B3-V-Senti. 2 #7	170	EVI Senti. 2 #6
33	NDWI #3	79	NDVI Senti. 2 #8	125	B4-V-Senti. 2 #7	171	NIRv Senti. 2 #6
34	Albedo #3	80	NDVI Senti. 2 #9	126	B1-V-Senti. 2 #8	172	EVI Senti. 2 #7
35	Temp1 #3	81	NDVI Senti. 2 #10	127	B2-V-Senti. 2 #8	173	NIRv Senti. 2 #7
36	Temp2 #3	82	B1-M-Senti. 2 #4	128	B3-V-Senti. 2 #8	174	EVI Senti. 2 #8
37	B1 Senti. 2 #4	83	B2-M-Senti. 2 #4	129	B4-V-Senti. 2 #8	175	NIRv Senti. 2 #8
38	B2 Senti. 2 #4	84	B3-M-Senti. 2 #4	130	B1-V-Senti. 2 #9	176	EVI Senti. 2 #9
39	B3 Senti. 2 #4	85	B4-M-Senti. 2 #4	131	B2-V-Senti. 2 #9	177	NIRv Senti. 2 #9
40	B4 Senti. 2 #4	86	B1-M-Senti. 2 #5	132	B3-V-Senti. 2 #9	178	EVI Senti. 2 #10
41	B1 Senti. 2 #5	87	B2-M-Senti. 2 #5	133	B4-V-Senti. 2 #9	179	NIRv Senti. 2 #10
42	B2 Senti. 2 #5	88	B3-M-Senti. 2 #5	134	B1-V-Senti. 2 #10	180	Senti. DPSVI-#11
43	B3 Senti. 2 #5	89	B4-M-Senti. 2 #5	135	B2-V-Senti. 2 #10	181	Senti. DPSVI-#12
44	B4 Senti. 2 #5	90	B1-M-Senti. 2 #6	136	B3-V-Senti. 2 #10	182	Senti. DPSVI-#13
45	B1 Senti. 2 #6	91	B2-M-Senti. 2 #6	137	B4-V-Senti. 2 #10	183	Senti. DPSVI-#14
46	B2 Senti. 2 #6	92	B3-M-Senti. 2 #6	138	B1-E-Senti. 2 #4	184	VBF-10

**Table 4 sensors-22-08942-t004:** Features used in each classifier, as determined though our analysis in order to maximize classification accuracy. Index refers to the image index from Table 3.

Classifier #1		Classifier #2		Classifier #3	
Index	Name	Index	Name	Index	Feature Name
1	B1 Reflect. #1	180	Senti. DPSVI-#11	65	Senti. VV-#11
2	B2 Reflect. #1	181	Senti. DPSVI-#12	66	Senti. VH-#11
3	B3 Reflect. #1	182	Senti. DPSVI-#13	67	Senti. VV-#12
4	B4 Reflect. #1	183	Senti. DPSVI-#14	68	Senti. VH-#12
7	B7 Reflect. #1	184	VBF-10	69	Senti. VV-#13
15	B3 Reflect. #2	21	NDWI #2	70	Senti. VH-#13
16	B4 Reflect. #2	33	NDWI #3	71	Senti. VV-#14
19	B7 Reflect. #2			72	Senti. VH-#14
20	NDVI #2			73	DEM
184	VBF			92	B3-M-Senti. 2 #6
23	Temp1 #2			108	B3-M-Senti. 2 #10
127	B2 -V- Senti.2 #8			115	B2-V-Senti. 2 #5
				123	B2-V-Senti. 2 #7

**Table 5 sensors-22-08942-t005:** Confusion matrix of the traditional method (Classifier #1). Rows represent the classification, while columns represent the reference.

	Fen	Bog	Marsh	Swamp	Upland	Producer Accuracy	User Accuracy
Fen	134,845	2212	1753	13,691	3601	0.8638	0.7354
Bog	948	13,186	0	277	68	0.9106	0.7505
Marsh	1502	1	21,182	468	263	0.9045	0.7595
Swamp	721	40	117	8100	173	0.8851	0.1422
Upland	45,327	2130	4836	34,388	549,760	0.8638	0.9925
Overall accuracy						0.875	

**Table 6 sensors-22-08942-t006:** Confusion matrix of the proposed ensemble classifier. Rows represent the classification, while columns represent the reference.

	Fen	Bog	Marsh	Swamp	Upland	Producer Accuracy	User Accuracy
Fen	145,390	2132	1674	5508	1398	0.9315	0.8267
Bog	668	13,628	0	175	8	0.9412	0.8234
Marsh	822	0	22,200	290	104	0.9481	0.8142
Swamp	534	24	83	8462	48	0.9247	0.3203
Upland	28,445	766	3310	11,986	591,934	0.9301	0.9974
Overall accuracy						0.935	

**Table 7 sensors-22-08942-t007:** Number of the misclassified pixels with high confidence and their land cover assignments for the traditional and proposed methods.

	Fen	Bog	Marsh	Swamp	Upland
High conf. misclassified Pixels—Classifier #1	9714	604	321	414	20,167
High conf. misclassified Pixels—the proposed method	6608	505	265	348	15,370

**Table 8 sensors-22-08942-t008:** Matrix showing the assignments using the proposed ensemble classifier in comparison with Classifier #1 (the traditional method) for all pixels with changes in class assignment. Columns are the land covers that a misclassified pixel was first assigned to from Classifier #1. The rows correspond to the land cover that a pixel was assigned to by the ensemble classifier.

Final Land Cover						
		Fen	Bog	Marsh	Swamp	Upland
Initial Land cover	Fen	0	9	11	60	4918
	Bog	0	0	0	1	204
	Marsh	0	0	0	2	336
	Swamp	15,394	868	614	0	18,180
	Upland	1206	69	134	43	0

**Table 9 sensors-22-08942-t009:** Matrix showing pixel assignments by the proposed ensemble classifier in comparison with Classifier #1 (the traditional method) for a subset of the pixels shown in Table 7. For the pixels shown here, the correct class had the second largest support from the evidence, but the largest and second largest mass functions were similar. Columns are the land covers that a misclassified pixel was first assigned to from Classifier #1. The rows correspond to the land cover that a pixel was assigned to by the ensemble classifier.

Final Land Cover						
		Fen	Bog	Marsh	Swamp	Upland
Initial Land cover	Fen	0	1	2	28	4903
	Bog	0	0	0	0	204
	Marsh	0	0	0	2	333
	Swamp	7125	114	182	0	17,482
	Upland	708	22	34	10	0

**Table 10 sensors-22-08942-t010:** Land-cover breakdown of the misclassified pixels where two classifiers disagreed or all disagreed.

	Fen	Bog	Marsh	Swamp	Upland
Two Disagree	6817	150	235	523	27,402
All Disagree	752	8	76	1	7330

## Data Availability

The data that support the findings of this study are available from the corresponding author (A.J.) upon reasonable request.

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
