# Peer review of "An Advanced Data Fusion Method to Improve Wetland Classification Using Multi-Source Remotely Sensed Data"

_sensors, 2022, doi:10.3390/s22228942_

Round 1

Reviewer 1 Report

I am very interested in the scientific problem that the article aims to solve. Moreover, the study should be a further extension of the authors' previous research. That's something I really appreciate. At present, I think the article still has some details to be improved.

1. The text in Figure 1 is too small to read clearly. The author can try to recreate the picture and delete unimportant information.

2. Page 5-6, Line 157-179, these paragraphs are more about the popularization of wetland science, is it really necessary to make such a detailed statement? I hope the author will reconsider and shorten this part.

3. In Table 1, the number of verification samples and training samples are not allocated in proportion as the author said, and even the number of verification samples in Upland is much larger than the number of training samples. I hope the author can explain this.

4. The title of Page 15, Line 459, Table 4 is incomplete. The text after 'from' is missing.

5. Figure 5 shows that the research has achieved satisfactory results. However, the text in the picture is too small to read. If the text is important, place it so the reader can see it.

6. Page 19, Line 563, the discussion part is written in detail. However, can the authors write a separate discussion of the advantages and disadvantages of this study? In addition, I also hope that the author can write a detailed research prospect in the discussion of advantages and disadvantages.

7. Page 24, Line 720, let's just put "Conclusions" in the title. The first paragraph is a detailed and repeated description of the research results. Readers need a short and powerful summary. The author can briefly and forcefully summarize the innovation and applicability of the research, and briefly and forcefully explain the further research.

Author Response

Response to Reviewer #1

 We would like to thank this reviewer for the valuable comments/suggestions. The following are points-to-points responses.

  1. The text in Figure 1 is too small to read clearly. The author can try to recreate the picture and delete unimportant information.

The Figure has been edited for better clarity and to remove unnecessary information.

  1. Page 5-6, Line 157-179, these paragraphs are more about the popularization of wetland science, is it really necessary to make such a detailed statement? I hope the author will reconsider and shorten this part.

We would like to thank the reviewer for this comment. We have edited the section to be come concise and to focus on the properties of the landcovers which relate best to the remote sensing aspect of the paper. The section now spans from Page 4-6 and Lines 158-201. It should be noted however that movement of the figures has changed the span of this section, but it has been reduced.

  1. In Table 1, the number of verification samples and training samples are not allocated in proportion as the author said, and even the number of verification samples in Upland is much larger than the number of training samples. I hope the author can explain this.

 We would like to thank the reviewer for catching this discrepancy. Upon closer examination we found  that there was a transcription error for the number of Upland training sample. It should be 2314364 (rather than 231436). This corresponds to a 78% to 22% split in training vs validation samples. Change was made in Table 1. In addition, the average proportion should be about 64% (instead of 60% reported in the original manuscript) and it was corrected as well. It now read “On average approximately 64% of the identified pixels were used for training and the rest for validation”

  1. The title of Page 15, Line 459, Table 4 is incomplete. The text after 'from' is missing.

 This has been corrected.

  1. Figure 5 shows that the research has achieved satisfactory results. However, the text in the picture is too small to read. If the text is important, place it so the reader can see it.

 The Figure has been edited for better clarity, as well as figure 6.

  1. Page 19, Line 563, the discussion part is written in detail. However, can the authors write a separate discussion of the advantages and disadvantages of this study? In addition, I also hope that the author can write a detailed research prospect in the discussion of advantages and disadvantages.

We would like to thank the reviewer for this comment. We have edited the discussion section regarding the proposed Ensemble Classifier to be structured in a more fluid way and to also more explicitly mention the advantage and disadvantages of the classifier. We hope that this meets the intent and spirit of the reviewer’s comment.

  1. Page 24, Line 720, let's just put "Conclusions" in the title. The first paragraph is a detailed and repeated description of the research results. Readers need a short and powerful summary. The author can briefly and forcefully summarize the innovation and applicability of the research, and briefly and forcefully explain the further research.

We would like to thank the reviewer for this comment. We have edited the section to be structured in a way to express the key findings and features of the study more succinctly and explicitly and what the next steps are. We have decided to leave some of the summary aspects of the conclusions in place as the  nature of the combined classes we used in classifiers 2 and 3 are part of the innovation and in order to contrast and explain that we also needed to leave in details about classifier 1. Also, the feature selection aspect of the classifiers, we feel is needed to be summarized and repeated again as its context is required.

Reviewer 2 Report

Based on multi-source remote sensing data, this paper uses the method of data fusion to solve the problems of low wetland classification accuracy and misclassification. Although it is innovative, there are many problems that need to be solved:

1. In the Abstract, there are too many background knowledge and conclusions, which can be summarized and condensed, and there are few method overviews, which can be appropriately expanded.

2. In the Introduction, there is a lack of introduction to state-of-the-art methods, such as deep learning for wetland classification, and remote sensing data fusion. In addition, on line 75, the paper mentions that the data fusion method is divided into feature-level and decision-level, should it also include pixel-level? Here are some newer references:  10.1109/TGRS.2022.3146296, doi.org/10.3390/rs14143492.

3. Add a paragraph at the end of the introduction with the structure of the paper. “The rest of the paper is structured as follows. In section 2, study area and images used are shown .....”

4. The Figures in this article should be drawn in the form of vector diagrams, such as PDF or EPS. Also, can Figure 1 be combined with Figure 2 into one Figure?

5. In the Study Area and Images Used, the different landcover classes have been introduced too much, and they are all from Ref. [47], which should be summarized.

6. In the overall framework Figure 4, the D-S theory, the key content of this paper, is not well reflected. And are classifiers 1-3 in a parallel relationship? Are classifiers 1-3 different classification methods, or do they all represent RF classifier that are just applied in different feature selection stages?

7. In the Results, this paper verifies the accuracy of the feature fusion results, and the results are effective, but there is a lack of comparison with other methods, such as SVM, CNN, etc.

8. What is the full name of ABMI in line 169 in the paper? Please complete and check the full text carefully.

Author Response

Response to Reviewer #2

 We would like to thank this reviewer for the valuable comments/suggestions. The following are points-to-points responses.

  1. In the Abstract,there are too many background knowledge and conclusions, which can be summarized and condensed, and there are few method overviews, which can be appropriately expanded.

We would like to thank the reviewer for this comment. We have since edited the abstract to be more succinct when it comes to the background knowledge and conclusions and have expanded out some of the methodology, while maintaining the word count.

  1. In the Introduction, there is a lack of introduction to state-of-the-art methods, such as deep learning for wetland classification, and remote sensing data fusion.In addition, on line 75, the paper mentions that the data fusion method is divided into feature-level and decision-level, should it also include pixel-level? Here are some newer references:  10.1109/TGRS.2022.3146296, doi.org/10.3390/rs14143492.

We appreciated the comments from this reviewer. Regarding pixel-level fusion, it was not initially included, considering that multi-source data was not explicitly analyzed and potentially not fully utilized in the context of cover type classification. In the revised manuscript, several sentences were added to explain this (lines 74-82).  In the revised manuscript, we also reviewed deep learning methods in the context of classification using multi-source remotely sensed data (lines 131-141)

  1. Add a paragraph at the end of the introduction with the structure of the paper. “The rest of the paper is structured as follows. In section 2, study area and images used are shown .....”

We would like to thank the reviewer for this comment. We have added a paragraph at the end of the introduction to address the reviewer’s concerns. This paragraph spans from lines 142-146.

  1. The Figures in this article should be drawn in the form of vector diagrams, such as PDF or EPS.Also, can Figure 1 be combined with Figure 2 into one Figure?

Figures have been edited to be clearer and figure 1 now is a combination of figure 1 and previously figure 2.

  1. In the Study Area and Images Used, the different landcover classes have been introduced too much, and they are all from Ref. [47], which should be summarized.

We would like to thank the reviewer for this comment. We have edited the section to be come concise and to focus on the properties of the landcovers which relate best to the remote sensing aspect of the paper. The section now spans from Page 4-6 and Lines 158-201. It should be noted however that movement of the figures has changed the span of this section, but it has been reduced.

  1. In the overall framework Figure 4, the D-S theory, the key content of this paper, is not well reflected.And are classifiers 1-3 in a parallel relationship? Are classifiers 1-3 different classification methods, or do they all represent RF classifier that are just applied in different feature selection stages?

We would like to thank the reviewer for this comment. The classifiers 1-3 work with one another in a parallel relationship, where they classify their respective land cover classes in parallel and are then combined using D-S Theory. The classifiers themselves classify different class types or combinations thereof, and are all executed using a RF classifier. We have since edited the section to be clearer and we hope this addresses the reviewers concerns. Also, in the edited manuscript Figure 4 is now Figure 3 as we have combined Figure 1 and 2.

  1. In the Results, this paper verifies the accuracy of the feature fusion results, and the results are effective, but there is a lack of comparison with other methods, such as SVM, CNN, etc.

We would like to thank the reviewer for this comment and to express to the reviewer that we have done comprehensive experiments with SVM, K-NN, and Naive-bayes classifiers using this framework. In general we found that the classification accuracies was some 5-8% less when compared to the results from RF. For brevity and focus we chose to only include results from RF. However, in the discussion we now also mention some of those results and the larger context of those explorations in section 5.4.

  1. What is the full name of ABMI in line 169 in the paper? Please complete and check the full text carefully.

Alberta Biodiversity Monitoring Institute – language and text has been edited.

Round 2

Reviewer 1 Report

  • The author has revised the article according to my comments.

Reviewer 2 Report

The authors carefully revised the paper, and the quality has been greatly improved. I think the paper can be published now.